# Teaching an Old Molecule New Tricks: Drug Repositioning for Duchenne Muscular Dystrophy

**DOI:** 10.3390/ijms20236053

**Published:** 2019-11-30

**Authors:** Libero Vitiello, Lucia Tibaudo, Elena Pegoraro, Luca Bello, Marcella Canton

**Affiliations:** 1Department of Biology, University of Padova, via U. Bassi 58/B, 35131 Padova, Italy; libero.vitiello@unipd.it; 2Interuniversity Institute of Myology (IIM), Administrative headquarters University of Perugia, Piazza Lucio Severi 1, 06132, Perugia, Italy; lucia.tibaudo@unipd.it; 3Department of Biomedical Sciences, University of Padova, via U. Bassi 58/B, 35131 Padova, Italy; 4Department of Neurosciences, University of Padova, Via Giustiniani, 5-35128 Padova, Italy; elena.pegoraro@unipd.it; 5Fondazione Istituto di Ricerca Pediatrica Città della Speranza-IRP, Corso Stati Uniti, 4, 35127 Padova, Italy

**Keywords:** Duchenne muscular dystrophy, drug repurposing, druggable targets, clinical use, mdx mice

## Abstract

Duchenne muscular dystrophy (DMD) is one of the most severe forms of inherited muscular dystrophies. The disease is caused by the lack of dystrophin, a structurally essential protein; hence, a definitive cure would necessarily have to pass through some form of gene and/or cell therapy. Cell- and genetic-based therapeutics for DMD have been explored since the 1990s; recently, two of the latter have been approved for clinical use, but their efficacy is still very low. In parallel, there have been great ongoing efforts aimed at targeting the downstream pathogenic effects of dystrophin deficiency using classical pharmacological approaches, with synthetic or biological molecules. However, as it is always the case with rare diseases, R&D costs for new drugs can represent a major hurdle for researchers and patients alike. This problem can be greatly alleviated by experimenting the use of molecules that had originally been developed for different conditions, a process known as drug repurposing or drug repositioning. In this review, we will describe the state of the art of such an approach for DMD, both in the context of clinical trials and pre-clinical studies.

## 1. Introduction

Duchenne muscular dystrophy (DMD [MIM: 310200]) is an X-linked, progressive form of inherited muscular dystrophies, with an incidence of about 1:5000 [1,2]. The disease is caused by lack of the dystrophin protein, due to mutations of the DMD gene. Dystrophin is a cytoskeletal component that plays an essential role in maintaining muscle fiber membrane integrity, by connecting the fiber’s contractile apparatus to the sarcolemma and ultimately to the external extracellular matrix. Its deficiency results in dramatic muscle deterioration, with repeated cycles of fibers’ degeneration/regeneration that eventually result in fibro-adipose substitution of the muscle tissue. Patients present a clear histopathological phenotype already at birth, but clinical symptoms, in the form of delayed motor milestones, usually appear at age 3–5. The disease is invariably progressive and DMD boys are usually wheelchair-bound by age 10–13. Thanks to a steady improvement in palliative care, chiefly nocturnal assisted ventilation, life quality and expectancy have increased, but most patients eventually succumb due to respiratory/cardiac failure by the third decade [3].

Given that DMD is caused by the lack of a structurally essential protein, a definitive cure would necessarily have to pass through some form of gene and/or cell therapy. During the past two decades, many promising therapeutic approaches have been developed and tested in DMD animal models, but so far, clinical trials in patients have led to much less impressive results. Two novel ‘genetic-based’ drugs have recently been approved for clinical use by either Food and Drug Administration (FDA) or European Medicines Agency (EMA), but they are aimed at specific subsets of patients and their efficacy is still subjected to debate. For instance, at the time of this writing, exon 51 skipping drug Eteplirsen has been approved for commercialization by FDA but not by EMA, and it is potentially applicable to less than 15% of the DMD population [4]. DMD is a complex disease, whose management requires a multi-disciplinary approach [5]. When it comes to pharmacological management, the only available options capable of delaying the progression of the disease in terms of skeletal muscle function are two glucocorticoids, deflazacort or prednisone, which to this day represent the gold standard for DMD in terms of pharmacological therapy [6]. Both drugs can significantly prolong ambulation and preserve muscle force in most DMD patients, albeit at the cost of heavy side effects. In this situation, the search for ‘druggable’ targets is of paramount importance in DMD research and is indeed the subject of intense investigation, with almost 50 active clinical trials with various molecules at the time of writing (https://clinicaltrials.gov/).

DMD belongs to the category of the rare diseases (a.k.a. “orphan disease”), defined by a prevalence below 1:2000 in Europe and of less than 200,000 patients in the USA [7]. The quest for a cure for rare diseases is hampered by the fact that the high costs of basic research and subsequent clinical trials have to be faced in the front of a relatively small revenue potential for the pharmaceutical companies. Part of the burden, especially for basic research, is often sustained by patients’ Associations, but these can rarely afford the costs of development and clinical testing for new drugs. For these reasons, when it comes to rare diseases, a cost- and time-effective strategy relies on drug repurposing, i.e., the finding of new indications for a drug that has already been approved for a different condition [8]. This allows for skipping most, if not all, pharmacokinetics and safety studies, greatly speeding up the process of assessing actual efficacy in patients. Such an approach is based on the consideration that rare and common diseases often share several clinical features, and that the similarities in many cases are also reflected at the level of pathogenetic mechanisms [9]. Indeed, even in the case of classic Mendelian diseases, the pathological phenotype is rarely linked solely to the (mal)function of a single gene product, but rather derives from the alteration of multiple molecular networks [10]. This consideration, compounded with the fact that in most cases drugs interact with multiple targets besides the intended ones [11], explains why a molecule originally designed for a specific disease could also prove useful in a different one.

In this review, we will summarize the state of the art of drug repurposing for DMD, both at the level of pre-clinical studies in dystrophic (*mdx*) mice and of clinical trials, taking into account only molecules that have been already approved for commercialization. The drugs are summarized in Table 1 and their structural formulas in Figure 1.

## 2. The Gold Standard: Glucocorticoid Corticosteroids

Glucocorticoid corticosteroids (GCs) are potent anti-inflammatory agents first isolated from adrenal cortex in the 1940s, and initially adopted with exceptional results in rheumatoid arthritis. This discovery resulted in the Nobel Prize in Physiology and Medicine being awarded in 1950 to Kendall, Reichstein, and Hench—one of the shortest periods of time occurring between any discovery and the attribution of a Nobel Prize [12].

One of the main oral forms of GCs, prednisolone or its pro-drug prednisone, was first used to treat DMD in the early 1970s, with clear evidence of efficacy even in a small seminal study of 14 patients by Drachmann and Colleagues [13]. This may be considered the original, and so far, the most successful example of drug repurposing for DMD (Table 1).

Since then, several other studies have been conducted, as effectively summarized in a recent Cochrane review [14]. It is demonstrated beyond doubt that glucocorticoids are effective in improving and stabilizing skeletal muscle strength in DMD over a time period of one to two years. The benefit in preserving motor function in the long term is also well established (e.g., prolonged ambulation by approximately 2–3 years), but the evidence is weaker as it is mostly based on observational studies [15,16] rather than randomized controlled trials. 

Side effects of glucocorticoids, especially occurring with long-term treatment, are well known across pediatric and adult populations. Different from adults, in which hypertension and glucose intolerance are frequent, in young boys the main issues are weight gain, behavioral disturbances, growth stunting, and osteoporosis [6]. Gastrointestinal symptoms and cataracts are also quite common. Virtually all DMD patients who undergo prolonged GC treatment according to standards of care [17] experience some degree of GC-related adverse effects. However, these are usually manageable and must be proactively screened for, prevented, and counteracted, tailoring dose on the individual benefit/risk profile. Nevertheless, there is clearly a dire need for safer treatment options in DMD.

Several treatment regimens (daily, alternate day, 10 days on and 10 days off, etc.) have been experimented and applied in the clinic, using both prednisone/prednisolone and deflazacort, and with few randomized studies comparing the relative efficacy and safety of different regimens [18]. A global, National Institutes of Health-sponsored, randomized clinical trial comparing daily prednisone, daily deflazacort, and 10-days-on-10-days-off prednisone named finding the optimum corticosteroid regimen for DMD (FOR-DMD) is currently reaching its completion in the U.S. and several European countries.

Despite lingering uncertainties about the optimum regimen, GC treatment is unanimously considered useful and recommended as a standard of care in DMD [6,17]. Treatment is usually started around the age of 4 to 7 years, at the dose of 0.75 mg/kg/day prednisone or 0.9 mg/kg/day deflazacort, and is adjusted according to individual response and adverse effects (osteoporosis, weight gain, cataracts, etc.). Increasingly, treatment is protracted beyond loss of independent ambulation into the early and late non-ambulatory stages of DMD, because of evidence from observational studies of beneficial long-term effects on upper limb function, lung function, and efficacy in preventing severe scoliosis [16,19].

The mechanisms of action of glucocorticoids in dystrophinopathy are complex and manifold. The potent inhibition of nuclear actor kappa-B (NF-κB) and its downstream pro-inflammatory effects are surely crucial mechanisms (Figure 2), but other relevant pathways also include fiber type transition from glycolytic to oxidative, widespread regulation of gene expression, membrane stabilization, stimulation of regeneration and repair, regulation of calcium metabolism, and possibly regulation of utrophin expression [20,21,22,23].

## 3. Simvastatin

Simvastatin is a lipid-lowering agent widely used in clinics for treating high blood LDL cholesterol levels and associated cardiovascular diseases. It derives synthetically from a fermentation product of the fungus *Aspergillus terreus*. Hydrolyzed in vivo to an active metabolite, simvastatin competitively inhibits hepatic hydroxymethyl-glutaryl coenzyme A (HMG-CoA) reductase, the enzyme that catalyzes the conversion of HMG-CoA to mevalonate, a key step in cholesterol synthesis. Besides these cholesterol lowering effects, statins also reduce inflammation, oxidative stress, and fibrosis through a cholesterol-independent mechanism [36,37,38].

Recently, a preclinical study in *mdx* mice showed that simvastatin reduced muscle damage and enhanced muscle function, by reducing inflammation, oxidative stress, and fibrosis [39]. Further analyses also showed positive effects on cardiac function in the same murine model [39,40]. More preclinical experimentations are underway to help better characterize the risks and benefits of statins in DMD and inform the optimal molecule to move into clinical studies.

## 4. N-acetylcysteine and Antioxidants

N-acetyl cysteine has been approved by FDA as the mainstay of therapy for acetaminophen toxicity, as it is highly effective in the treatment of potentially hepatotoxic overdoses. It is also approved for diseases associated with excessive, viscous mucous secretions such as pneumonia, bronchitis, and cystic fibrosis. The main molecular mechanism is due to its ability to replete glutathione reserves by providing cysteine, which is an essential precursor in glutathione synthesis. Glutathione, in its reduced form, is a crucial antioxidant by itself and also a substrate for different antioxidant enzymes [50]. In case of significant depletion of glutathione, N-acetyl cysteine also acts as a direct antioxidant, as a thiol compound. The use of N-acetylcysteine in *mdx* mice has been found to alleviate skeletal muscle dysfunction and pathologic histology [51]. Similar results were observed by treating *mdx* mice with another antioxidant, (−)-epigallocatechin gallate, the major polyphenolic component of green tea extract, [52]; this molecule has also been used in a recently completed DMD clinical trial (NCT01183767), for which no results have yet been published. However, the use of non-selective antioxidants is quite controversial, as recently discussed [64,65,66].

## 5. Safinamide and MAO Inhibitors

Oxidative stress and mitochondrial dysfunction are known to play a key role in DMD [55,67,68,69,70,71,72]. A crucial source of reactive oxygen species (ROS) in dystrophic muscles is monoamine oxidase (MAO) [55,56,72], a mitochondrial enzyme widely studied for its role in the central nervous system [57]. The two isoforms of MAO, A and B, are located in the outer mitochondrial membrane and catalyze the oxidative deamination of different biogenic amines to generate aldehydes and H_2_O_2_. Pathologic excess of H_2_O_2_ have been shown to be involved in the oxidation of contractile proteins both in ischemic heart and dystrophic skeletal muscle [56,72,73,74,75]. Consistently, treatment with pargyline, an inhibitor of both MAO-A and MAO-B, reduced tropomyosin oxidation and led to improvement of the dystrophic phenotype in *mdx* and *col6a1^−/−^* mice [72]. MAO has also been shown to be overactivated in myoblasts from patients with collagen VI myopathies and DMD [55,56]. More recently, novel and better tolerated inhibitors of the B isoform (MAO-Bi) have been introduced in the clinic for neurological disorders [76]. The advantage of inhibiting MAO-B is to avoid the risk of hypertensive crises, which is associated with inhibition of the MAO-A isoform. In addition, the molecular structure of MAO-B has been identified at high resolution [77,78,79], thus allowing the design of highly specific inhibitors. Among them, safinamide is a selective and reversible MAO-Bi, with an improved profile of efficacy and safety, that has been introduced in the market for Parkinson’s disease. In a recent report, Safinamide has been shown to markedly improve muscle function in *mdx* mice, as well as to reduce oxidative stress and mitochondrial dysfunction in muscle cells from DMD patients [56].

## 6. Sunitinib

Recently, Fontelonga and Colleagues have shown that sunitinib (SU11248), a multi receptor tyrosine kinase (RTK) inhibitor approved for the treatment of renal cell carcinoma [53] and gastrointestinal stromal tumors, provided benefits in *mdx* mice [54]. Treatment with this drug promoted satellite cell (SC) activation and myogenic regeneration, leading to significantly improved muscle disease pathology and functional skeletal muscle force production. Such effects have been linked to Sunitinib’s capability to act as a potent α7ß1 integrin enhancer, thereby stimulating satellite cell activation and increasing myofiber fusion via the STAT3 pathway [54].

## 7. Idebenone

Idebenone is a synthetic short-chain coenzyme Q10 analogue, which was initially developed for the treatment of degenerative neurocognitive disorders, but in this indication it never reached significant success. However, it has been approved by Europe Medicines Agency (EMA) for the treatment of Leber’s hereditary optic neuropathy. It transfers electrons directly to the mitochondrial complex III of the respiratory chain, thereby restoring ATP intracellular levels, and also acts as an antioxidant and a free radical scavenger. As discussed above, there is a strong rationale for treatment with respiratory chain co-factors and antioxidants in a disease, such as DMD, whose pathophysiology is characterized by a cellular energetic failure and chronic oxidative stress [80].

As idebenone had been applied in the treatment of the cardiological complications of a different neurogenetic disease, Friedreich’s Ataxia [81,82], it does not come as a surprise that the first clinical trials of idebenone in the DMD field were targeted at treating dilated cardiomyopathy. An early, phase IIa “pilot” study compared 13 DMD patients aged 8–16 years, treated with 450 mg/day idebenone, with eight matched controls [58] with a 12-month follow-up (DELPHI trial). Echocardiographic measures of left ventricular contractility showed a positive trend but no significant treatment-related differences; however, a significant difference between treated and control groups was observed in peak expiratory flow (PEF) values. Subsequent trials then focused on respiratory endpoints, and primarily on GC-naïve patients who had shown maximum idebenone-related effects on PEF, according to post-hoc analyses of DELPHI data [59]. The phase 3 DELOS trial [60,61] therefore recruited 64 DMD patients aged 10 to 18 years and not on concomitant GCs, who were randomized 1:1 to receive a higher idebenone dose (900 mg/day divided into three doses) or placebo. As in previous phase II trials, idebenone was well tolerated, and it attenuated the decrease of PEF over 12 months by about 6% of age/height predicted values (*p* < 0.05). This positive trial established idebenone as a promising treatment option for DMD at several disease stages, including later non-ambulatory phases for which the availability of treatments is even scanter than for ambulatory patients. Some limitations included the relatively small sample size for a phase 3 trial and some age discrepancy between treated and placebo groups. Furthermore, at the time when these results were published in 2015, it was somewhat difficult to gauge the clinical significance of the 6% difference in PEF reduction, because of the lack of longitudinal “natural history” studies of pulmonary function tests (PFTs). This knowledge gap has recently been filled and the average decrease of PEF in DMD patients aged 10–18 seems to be around 5%/year [16]. Post-hoc analyses of DELOS data showed a reduced rate of respiratory complications [62] (primarily chest infections) and longer retainment of PFT values above thresholds indicating the need for non-invasive ventilation [63]. Currently, another global phase 3 study (SIDEROS) is ongoing, aiming to demonstrate idebenone efficacy also in GC-treated DMD patients.

## 8. Tamoxifen

Tamoxifen, a selective estrogen receptor modulator (SERM), is a one of the most commonly used drugs in the treatment of estrogen receptor-positive breast cancer. However, it is also known to possess numerous other therapeutically relevant effects, as it has been shown to act a ROS scavenger, an anti-apoptotic agent, and an inhibitor of fibroblast proliferation [41]. Starting from these considerations, Dorchies and Colleagues tested its use in the 5Cv strain of *mdx* mice, subjecting the animals to long-term treatment with the drug. This resulted in remarkable improvements of muscle force and reduction of fibrosis of the diaphragm and the heart [41]. Interestingly, the efficacy of tamoxifen was observed also in mice with another severe congenital muscular disorder, the fatal X-linked myotubular myopathy [42].

Shifting from murine to human dystrophinopathy, in DMD patients the influence of sexual hormones has been demonstrated to be patho-physiologically relevant. Because of a combination of causes related to the disease itself and GC treatment, DMD patients often exhibit some degree of hypogonadism leading to pubertal delay [17], and their muscles overexpress estrogen receptors (ERs). Recently, tamoxifen has been granted an orphan drug designation for the treatment of DMD by the EMA, and a randomized, placebo-controlled phase 3 clinical trial with a multi-center design is has started to assess the efficacy of tamoxifen in DMD [43].

## 9. Metformin

Metformin belongs to the class of biguanides, a family of compounds with complex mechanisms of action, including mainly a regulation of respiratory chain activity and the AMP-activated protein kinase (AMPK) cascade, which ultimately inhibits gluconeogenesis from the liver [28]. Initially approved and widely used for its blood glucose lowering effects, metformin is a drug that has a multifaceted potential for repurposing because of its pleiotropic effects. For instance, it is also approved for the treatment of polycystic ovary syndrome, and its regulatory action on the mammalian target of rapamycin (mTOR) signaling pathway implies an interesting potential as an anti-cancer agent [29]. Preclinical studies of metformin in the *mdx* model of dystrophinopathy have suggested that its long-term beneficial effects on muscle fibrosis and strength, as assessed by muscle histology and functional tests, respectively, can be independent from metabolic and AMPK-related effects, suggesting the existence of alternative mechanisms [30]. Other researchers have identified, in treated *mdx* mice, the activation of well-characterized phenotype-modifying pathways, such as peroxisome proliferator-activated receptor (PPAR)γ coactivator-1α (PGC-1α) and the upregulation of utrophin [31].

The first application of metformin to human DMD in the literature is described in a trial that recruited a heterogeneous population of pediatric patients affected with conditions characterized by motor dysfunction [32] (also including neural tube defects and other neuromuscular conditions), and successfully showed a reduction of visceral adiposity and insulin resistance. However, later studies have focused on more disease-specific mechanisms of action of metformin in muscle dystropathology. For example, the ability of metformin to upregulate neuronal nitric oxide synthase (nNOS), which is defective in dystrophin-deficient muscle (see also the following paragraph), has provided a strong rationale for its association with nitric oxide (NO) precursors such as l-citrulline. After initial, small proof-of-concept studies of the metformin/l-citrulline association in Becker muscular dystrophy (BMD) and DMD [33,34], which showed promising results on muscle metabolism biomarkers and preliminary clinical evaluations, metformin/l-citrulline experimentation is being moved on to a larger, randomized, double-blind, placebo-controlled trial with a duration of six months and the clinically oriented motor function measure (MFM) as a primary outcome [35].

## 10. PDE5 Inhibitors and Nitrate Drugs

Phosphodiesterase type 5 inhibitors (PDE5i) are a class of molecules approved for the treatment of erectile disfunction and pulmonary hypertension, as blocking PDE5 leads to increased levels of cGMP and hence to the relaxation of smooth muscle cells within blood vessels. Starting from the observation that one of the pathological features of dystrophic muscle is the presence of functional ischemia (i.e., the incapability of achieving sufficient blood flow upon muscle activity), Asai and Colleagues treated *mdx* mice with tadalafil, reporting decreased fiber necrosis upon repeated tetanic contractions [44]. Shortly after, another group reported that the heightened fatigue response to mild exercise present in *mdx* mice was indeed due to impaired vascular adaptation upon muscle contraction, due to the loss of sarcolemma-localized nNOS and consequent impairment of cGMP-mediated vaso-modulation [45]. In the same report, Kobayashi et al. showed that treatment with a PDE5i could greatly improve blood flow and reduce edema in the muscles of *mdx* mice after exercise [45]. Subsequently, the group led by RG Victor tested the efficacy of administering the PDE5i tadalafil in improving functional muscle ischemia pilot trials in BMD and DMD patients [46]. Both studies yielded positive results, although the outcome parameter was limited to the measurement of blood flow in the forearm upon handgrip exercise. More recently, other preclinical studies reported that the use of tadalafil led to functional and histological improvements in skeletal muscle in *mdx* mice [47] and could delay the onset of cardiomyopathy both in *mdx* mice and in the golden retriever canine model of DMD [48]. The clinical development program of Tadalafil culminated in a large phase 3, double blind, placebo-controlled, randomized clinical trial, with a three-arm design: Placebo, 0.3 mg/kg/day tadalafil, and 0.6 mg/kg/day tadalafil [49]. Notably, this trial represented one of the largest recruitment efforts in the history of DMD trials, with 331 randomized participants. While the known safety profile of tadalafil was confirmed, with no additional safety concerns in the Duchenne population than in the general population, unfortunately both tadalafil doses failed to slow down the decline in walking ability of the patients, as measured by the primary outcome, the 6-min walk test (6MWT). Secondary measures including other motor function scales, measurements of respiratory and cardiac function, and patient-centered assessments of quality of life, showed no significant drug-related effect.

## 11. Food Supplements

Flavocoxid is a mixture of bioflavonoid of plant origin, that was originally approved in 2004 by the FDA as a “medical food” with anti-inflammatory properties. It was commercialized as an integration to the treatment of osteoarthritis. Flavocoxid has an antioxidant/anti-inflammatory effect, and an ability to reduce levels of NF-κB signaling. After promising preclinical studies in mice [83], flavocoxid entered human experimentation in the form of phase I study, whose results, reported only as a conference abstract, suggested good tolerability and decreased levels of circulating inflammatory cytokines [84]. However, in 2017 FDA revoked the marketing authorization after several cases of acute liver injuries were reported and no further experimentations were reported after that.

## 12. Aminoglycosides (for Stop Codon Readthrough)

About 10% to 15% of DMD cases are due to nonsense mutations, i.e., point mutation [85] (most usually a single nucleotide substitution) leading to a single premature termination codon to be inserted into the open reading frame (ORF). Different from more frequent frameshifting mutations, in which the ORF downstream of the mutation becomes “meaningless” to the advancing ribosome, and fraught with tens of stop codons, with nonsense mutations the downstream genetic information is preserved, and could be translated by the ribosome if the mutation itself could be surpassed. Usually, the presence of a nonsense mutation causes the ribosomal complex to halt and activate the nonsense mediated decay pathway [86], resulting in the degradation of the transcript and absence of protein expression. However, the ribosomal complex does have the ability, in the right molecular context, to “read through” the nonsense mutation by inserting a random amino acid and continuing downstream translation [87]; an ability that can be enhanced by specific drugs. Aminoglycosides, a class of antibiotics approved for the treatment of several bacterial infections, have been known to be able to suppress nonsense mutations in animal models. Starting from this consideration, Barton-Davis and Colleagues tested the efficacy of gentamicin in *mdx* mice, demonstrating partial dystrophin restoration and increased resistance to force loss upon eccentric contraction, a typical hallmark of dystrophic muscle [24]. Gentamicin treatment led to positive results also in exercised *mdx* mice, a model that better reproduces the pathologic features found in DMD patients [25]. Therefore, an attempt was made to use gentamicin to obtain full-length dystrophin re-expression in a small study of four DMD patients [83]. Unfortunately, this could not be demonstrated in the short duration of the two-week study. On the other hand, a different small-scale study on four individuals who were either ambulatory or had lost ambulation for no more than four months, showed dystrophin re-expression in muscle biopsies performed after gentamicin infusions, as shown by immunohistochemistry and immunoblot [26]. Interestingly, the effect was apparent only in those patients with the more permissive UGA stop codon, and not in one patient with the UAA codon. A few years later, the largest study on aminoglycosides in DMD was published [27]. In this multi-center trial, different cohorts were treated with gentamicin infusions: A cohort of 10 nonsense DMD patients and a “control” cohort of eight patients with frameshift mutations (in order to control for hypothetical “non-genetic” effects of gentamicin) received a 14-day course of treatment, while two additional cohorts of 12 and 4 nonsense DMD patients were treated weekly and twice weekly, respectively. Dystrophin increase evaluated by immunofluorescence was statistically significant in the six-month treatment group, and the data did not support a preferential response with the UGA codon as observed by Politano et al.; rather, those patients who had higher levels of dystrophin at baseline appeared to respond better. Subsequent to the publication of these data, concerns about oto- and nephrotoxicity hindered a larger clinical development of gentamicin in DMD. Interestingly, a recent study in *mdx* mice suggested that aminoglycosides could also be beneficial for DMD treatment in a different capacity, namely as facilitators of the delivery of morpholino antisense oligonucleotides to skeletal muscle [88]. As for the stop-codon readthrough strategy, in DMD this is now being pursued with the use of novel, non-aminoglycoside, synthetic molecules [89,90]. Of these, Ataluren [84,85] has progressed through all the phases of clinical trials and has recently been approved by EMA.

## 13. Cardiological Drugs (ACE Inhibitors, Beta-Blockers, Eplerenone)

Dilated cardiomyopathy (DCM) is one of the major complications of DMD, and one of the most relevant determinants of life expectancy [91], especially after the implementation of mechanical ventilation has significantly reduced and delayed death due to respiratory insufficiency. Several drugs are used in the clinical management of dystrophin-related DCM, and it could be debated whether this can be regarded as “repurposing”. In fact, several physio-pathological mechanisms are shared between dystrophin-related and more common (e.g., ischemic) cardiomyopathies: Fibrosis, dilated remodeling, increased oxygen consumption, and adrenergic hyperactivation being some of the most relevant. Therefore, cardiac drugs target these mechanisms in similar ways in DMD as in “common” cardiopathic patients. Here, we will only briefly review the main pharmacological categories employed in the treatment of dystrophin-related DCM, and the most important evidence supporting their use.

One of the main challenges in studying cardiological endpoints in DMD is that several years of observation are usually needed to observe a meaningful change in available measures of cardiac function, effectively summarized in a recent review of the literature [92], to which our readers are pointed for more in-depth discussion. The most widely used outcomes are echographic measures of left ventricular systolic function, such as fractional shortening and ejection fraction, as well as of dilation, e.g., left ventricular telediastolic volume normalized by body surface. Electrocardiographic alterations such as pseudo-necrotic Q waves are also relevant and often precede echographic changes. The recent advances in magnetic resonance imaging of the heart, which allow a very accurate characterization of myocardial fibrosis—revealed by late gadolinium enhancement—and provide very sensitive contractility measures such as myocardial strain estimations, have increased its role both in clinical management and research.

Angiotensin-converting enzyme inhibitors (ACEi) are probably the most frequently prescribed cardiac medications in DMD [93]. By inhibiting the renin-angiotensin pathway, they reduce peripheral circulatory resistance and blood volume (through a reduction of mineralocorticoid secretion). The hypothesis behind their application in DMD was that the subsequent reduction of afterload would release the myocardium from the mechanical stress that is known to cause cellular damage when dystrophin is lacking [94]. In fact, the ACEi perindopril did show an ability to delay the progression to a reduced systolic function of the left ventricle, in a randomized, blinded trial [95]. The effect of ACEi may also be due to their secondary anti-fibrotic properties, a consequence of the reduction of mineralocorticoids. Longitudinal follow-up studies collecting data up to 10 years of observation proved that perindopril is also able to prevent and delay the onset of DCM [96], and not just to slow down its progression once it has started. Currently, therefore, prophylactic treatment with ACEi is increasingly being implemented, starting from the age of around 10, even in DMD children with no evidence of DCM, although international standard-of-care guidelines leave this clinical choice at the discretion of cardiologists [93].

The second mainstay of DMD cardiac therapy are ß-blockers (BBs), which reduce hyperadrenergic activation, oxygen consumption, and the risk of dangerous arrhythmias started by foci of myocardial fibrosis [97]. Their use is recommended by current standards of care [43], especially in patients with marked tachycardia. The association with ACEi has shown good results both as treatment and prophylaxis of DCM [98], although randomized, controlled trials with significant primary endpoint results are lacking.

As mentioned above while discussing the effects of ACEi, the secretion of mineralocorticoids has detrimental consequences for the DMD heart, because of both afterload increase (i.e., salt retention) and the activation of pro-fibrotic pathways. Mineralocorticoid antagonists, also known as potassium-sparing diuretics, are effective in contrasting these mechanisms and have a place in the treatment of DMD DCM [93]. While the most common agent of this class was spironolactone, a recent study of a newer molecule, eplerenone [99], showed significantly improved left ventricular function in 22 treated DMD patients vs. 20 controls, on the background of treatment with either ACEi or BBs. A strength of this study was the use of sensitive, heart-MRI derived outcome measures such as left ventricular circumferential strain, which allowed to overcome the relatively small sample size and short observation time of 12 months.

Other agents (diuretics, inotropic drugs, etc.) may sporadically be prescribed; a newer drug combination, sacubitril/valsartan, has recently been shown to exhibit superior therapeutic efficiency compared to standard ACEi in heart failure patients, both adult and pediatric [100,101]. For this reason, its use has been suggested also in DMD patients, but at present, no specific studies have been reported [102]. Currently, though, the combination of ACEi, BBs, and mineralocorticoid antagonists, under the guidance of a cardiologist with experience in dystrophinopathies, represents the core of cardiac care for DMD, and clearly—despite the relative lack of high-class evidence, in the context of evidence-based cardiology—has the potential to significantly enhance life quality and expectancy in DMD [102].

## 14. Conclusions

Duchenne muscular dystrophy was the second genetic disease whose underlying molecular defect was discovered by the then novel “reverse genetics” approach, more than 30 years ago [103]. Since then, the intrinsic features of dystrophin (e.g., its large size and complex structural role, the multiple type of mutations found in patients) have hampered the many attempts at developing resolutive therapies. The fact that in the recent past two genetic-based therapies have been approved for commercialization offers hope for further positive developments, as do the indications that gene replacement therapy with specifically crafted micro-dystrophins appears to be providing positive results in clinical trials [104]. However, molecular therapies with mutation-specific drugs, such as exon skipping oligonucleotides or small molecules promoting stop codon readthrough, are intrinsically limited to restricted DMD sub-populations [92]. Furthermore, some patients may not be eligible for gene therapy because of pre-existing immunity to viral vectors and other reasons [92]. Last but not least, both exon skipping and micro-dystrophin expression would be of relatively little help for all those patients in whom the amount of actual muscle tissue has already been drastically reduced. In these latter cases, a cell transplantation approach would be the ideal solution, but attempts in this direction have so far failed and no clinical trials are presently ongoing. For all of these reasons, there is a definite and pressing need to develop pharmacological therapies capable of addressing the numerous downstream consequences of dystrophin deficiency, in order to improve life expectancy and quality of life in patients. This is why drug repurposing is playing an increasingly important role in the quest for new, efficacious and affordable therapies for DMD, and will increasingly do so in the foreseeable future.

## Figures and Tables

**Figure 1 ijms-20-06053-f001:**
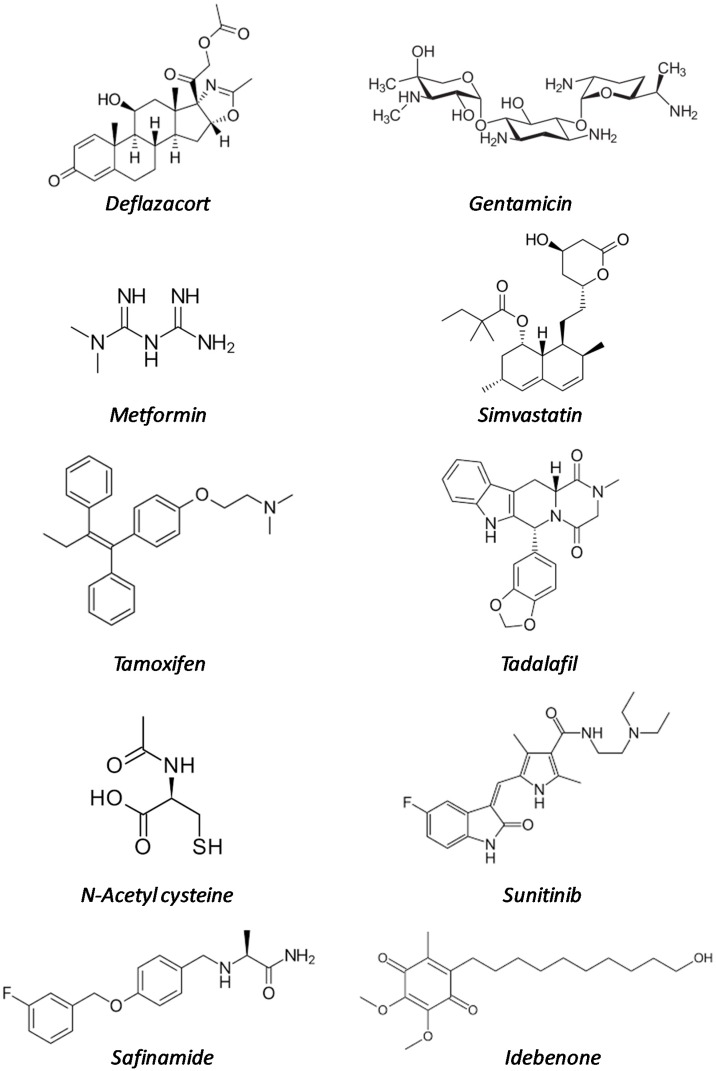
Structural formulas of the repurposed drugs cited in Table 1.

**Figure 2 ijms-20-06053-f002:**
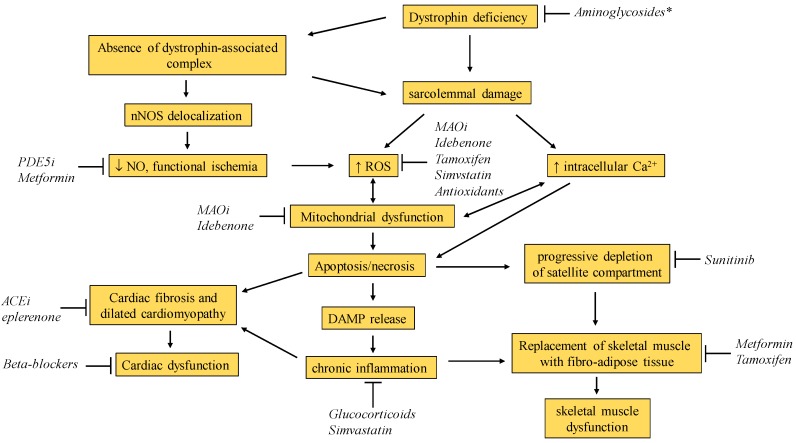
Schematic representation of pathogenic mechanisms in DMD. Blunted arrows indicate the hypothesized point of actions of repurposed drugs currently under investigation. NO, nitric oxide; nNOS, neuronal nitric oxide synthase; PDE5i, phosphodiesterase 5 inhibitors; ROS, reactive oxygen species; MAOi, Monoamine oxidase inhibitors; ACEi, angiotensin converting enzyme inhibitors; DAMP, damage-associated molecular patterns; * initially considered for stop codon readthrough, but now replaced by a specifically designed molecule.

**Table 1 ijms-20-06053-t001:** Examples of repurposed drugs that are being investigated for Duchenne muscular dystrophy (DMD) treatment.

Drug Name	Original Indication (Date of Approval)	Mode of Action	Development Phase in DMD	References
Deflazacort	Rheumatoid arthritis (1955)	Multiple mechanisms	Marketed	[13,14,15,16,17,18,19,20,21,22,23]
Gentamicin	Bacterial infections (1987)	Protein synthesis inhibitor	Clinical trials	[24,25,26,27]
Metformin	Type II diabetes (1995)	Multiple mechanisms	Clinical trials	[28,29,30,31,32,33,34,35]
Simvastatin	Familial hyperlipidaemia (1998)	HMG-CoA reductase inhibitor	Preclinical studies	[36,37,38,39,40]
Tamoxifen	ER-positive breast cancer (1998)	Estrogen receptor (ER) modulator	Clinical trials	[41,42,43]
Tadalafil	Erectile dysfunction (2003)	PDE5 inhibitor	Clinical trials	[44,45,46,47,48,49]
N-acetyl cysteine	Acetaminophen overdose (2004)	Antioxidant	Preclinical studies	[50,51,52]
Sunitinib	Renal cell carcinoma and gastrointestinal tumors (2006)	Multireceptor tyrosin kinase (RTK) inhibitor	Preclinical studies	[53,54]
Safinamide	Parkinson’s disease (2015)	Monoamine oxidase B inhibitor	Preclinical studies	[55,56,57]
Idebenone	Leber neuropathy (2015, not FDA)	coenzyme Q_10_ analogue	Preclinical studies	[58,59,60,61,62,63]

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
