# Peer review of "Teaching an Old Molecule New Tricks: Drug Repositioning for Duchenne Muscular Dystrophy"

_ijms, 2019, doi:10.3390/ijms20236053_

Round 1

Reviewer 1 Report

The paper "Teaching an old molecule new tricks: drug repositioning for Duchenne Muscular Dystrophy" by M. Canton and co-workers is a review concerning with Duchenne muscular dystrophy, a genetic disease caused by a lack of dystrophin and still without a definitive cure.

In the last few years, many efforts have been done in order to find drugs that could at least ameliorate the clinical picture of the patients.

Since the development of new drugs is expensive and time consuming, drug repositioning is a very common approach widely applied.  

In this review the authors describe the drugs selected by drug repositioning that are currently under investigation for DMD, in preclinical or clinical studies.

The data are clearly shown, the manuscript is clearly written and deserves to be published.

I would only suggest adding the structural formulas of the cited drugs (in a separate figure or as new column in table 1).

Author Response

We thank the Reviewer for his/her comments. We have followed the Reviewer’s suggestion and added the structural formulas of the cited drugs in a separate figure.

Reviewer 2 Report

The manuscript entitled “Teaching an old molecule new tricks: drug repositioning for Duchenne Muscular Dystrophy” by Vitiello et al. aimed to revised the most common drug therapies developed and used in other diseases to help to treat patients diagnosed with Duchenne muscular dystrophy (DMD). The aim of the revision manuscript is achieved, given to the Reader a broad overview of the current genetic and drug treatment available in the market. Although the intention of the revision is good and the approach is correct, the Authors must address several issues before final decision. The major concerns are the description of drugs’ mechanisms are superficial, superficial discussion on the final outcome of the clinical trials or basic researches’ results in relation with the drug or treatment used, there are several wrong citations, there are too many revisions cited, and finally, lacking some fundamental citations from original papers.

The detail of some of these problems cited above are mentioned below. However, the Authors should not limited themselves only to the points listed here as examples, but also improve other parts of the manuscript.

Line 28: The Authors should correct this information. Davies and Nowack 2006 does not state this information of the incidence of DMD. The Authors should take care of citing previous reviews, which does not correctly cite the original source. A strong recommendation is to cite original research.

Lines 37-40: Here again, the Authors cited another revision of the topic, but in this case, the reference from Bourke et al., 2018 is based on a collection of data from different clinical trials. The Authors should consider to rewrite this paragraph a) using original papers, b) describing in more detail the structure of the gene, c)  describing in more detail the structure of the protein and the protein associated to dystrophin. In order to understand the relevance of the treatments described here, the Authors should also consider writing the clinical description of DMD patients and the progression of the disease.

Line 45: The Authors should state that until to this date EMA has not approved the marketing authorization of Eteplirsen. 

Line 48: The Authors need to cite this information. "When it comes to generalized clinical practice, the only available option is to try and delay the progression of the disease by using either of two glucocorticoids, deflazacort or prednisone;"

Line 56: The Authors should correct the a.k.a. abbreviation.

Line 79: The Authors should be more careful to read the papers before cite the references.  Barnes 2016 does not cite the Nobel Prize won by Kendall, Reichstein and Hench for their discoveries relating to the hormones of the adrenal cortex, their structure and biological effects. It is relevant in a revision to have the correct citation.

Lines 94-97: The Authors should use a reference for this paragraph. "Side effects of glucocorticoids, especially occurring with long-term treatment, are well known across pediatric and adult populations. Different from adults, in which hypertension and glucose intolerance are frequent, in young boys the main issues are weight gain, behavioral disturbances, growth stunting, and osteoporosis. Gastrointestinal symptoms and cataracts are also quite common."

Line 267: The Authors should change "the Authors" for Kobayashi et al.

Line 286: If the Authors would like to propose a discussion around flavocoxid, even that isn't approved by FDA anymore, the Authors should consider to discuss flavocoxid in the DMD aspect, which is the aim of the revision. In this case, the Authors should include and discuss the following reference:

Exp Neurol. 2009 Dec;220(2):349-58. doi: 10.1016/j.expneurol.2009.09.015. Epub 2009 Sep 25.

Flavocoxid counteracts muscle necrosis and improves functional properties in mdx mice: a comparison study with methylprednisolone. Messina S1, Bitto A, Aguennouz M, Mazzeo A, Migliorato A, Polito F, Irrera N, Altavilla D, Vita GL, Russo M, Naro A, De Pasquale MG, Rizzuto E, Musarò A, Squadrito F, Vita G.

Line 295-313: The Authors should include a discussion of the data obtained in mdx cardiac and skeletal muscles using Aminoglycosides.  Some suggestions are:

Aminoglycoside antibiotics restore dystrophin function to skeletal muscles of mdx mice. Barton-Davis ER, Cordier L, Shoturma DI, Leland SE, Sweeney HL. J Clin Invest. 1999 Aug;104(4):375-81.

Aminoglycoside Enhances the Delivery of Antisense Morpholino Oligonucleotides In Vitro and in mdx Mice. Wang M, Wu B, Shah SN, Lu P, Lu Q. Mol Ther Nucleic Acids. 2019 Jun 7;16:663-674. doi: 10.1016/j.omtn.2019.04.023. Epub 2019 May 2.

The Authors should also include in the discussion other reading through drugs such as RTC13.

Read-through compound 13 restores dystrophin expression and improves muscle function in the mdx mouse model for Duchenne muscular dystrophy. Kayali R, Ku JM, Khitrov G, Jung ME, Prikhodko O, Bertoni C. Hum Mol Genet. 2012 Sep 15;21(18):4007-20. doi: 10.1093/hmg/dds223. Epub 2012 Jun 12.

Lines 315-322: In order to understand the relevance of the drugs discussed in this chapter, the Authors should include clinical parameters that are influenced or modified in the DCM, such as echo, ECK and histological parameters.

Line 326: The Authors should include a reference for this statement. “Angiotensin-converting enzyme inhibitors (ACEi) are probably the most frequently prescribed cardiac medications in DMD. By inhibiting the renin-angiotensin pathway, they reduce peripheral circulatory resistance and blood volume (through a reduction of mineralocorticoid secretion).”

Line 333: The Authors should keep the abbreviation initiated on line 326 (ACEi).

Line339: The plural of ACEI is already included in the abbreviation and it is not necessary to add another "s"

Line 359: The Authors should provide data to support the usefulness of sacubitril/ valsartan.

Author Response

Reviewer 2

The manuscript entitled “Teaching an old molecule new tricks: drug repositioning for Duchenne Muscular Dystrophy” by Vitiello et al. aimed to revised the most common drug therapies developed and used in other diseases to help to treat patients diagnosed with Duchenne muscular dystrophy (DMD). The aim of the revision manuscript is achieved, given to the Reader a broad overview of the current genetic and drug treatment available in the market. Although the intention of the revision is good and the approach is correct, the Authors must address several issues before final decision. The major concerns are the description of drugs’ mechanisms are superficial, superficial discussion on the final outcome of the clinical trials or basic researches’ results in relation with the drug or treatment used, there are several wrong citations, there are too many revisions cited, and finally, lacking some fundamental citations from original papers.

We thank the Reviewer for his/her comments, and we appreciated his/her observation about the importance of citing “first hand” the most relevant original research. In drafting the manuscript, we tried to limit citations of other reviews only to introductory paragraphs about broad general concepts (e.g. DMD epidemiology or natural history), or concepts that were peripheral to the focus of this review. Conversely, we paid attention to cite all relevant, original papers when discussing the central topic of this review, e.g. repurposed drugs and their preclinical/clinical development. In this revised version, we made an additional effort to apply these criteria more thoroughly and hence have added several citations and replaced some others.

Importantly, in the revised manuscript we also provided more information in the drug mode of action.

Line 28: The Authors should correct this information. Davies and Nowack 2006 does not state this information of the incidence of DMD. The Authors should take care of citing previous reviews, which does not correctly cite the original source. A strong recommendation is to cite original research.

Done

Lines 37-40: Here again, the Authors cited another revision of the topic, but in this case, the reference from Bourke et al., 2018 is based on a collection of data from different clinical trials. The Authors should consider to rewrite this paragraph a) using original papers, b) describing in more detail the structure of the gene, c) describing in more detail the structure of the protein and the protein associated to dystrophin. In order to understand the relevance of the treatments described here, the Authors should also consider writing the clinical description of DMD patients and the progression of the disease.

In this case we do feel that our choice of reference is appropriate. Specifically, we believe that providing a thorough description of the genetic and molecular basis of DMD, one of the best studied monogenic diseases and also one with the most complex phenotype, would require a review in itself and would hence be off-target with respect to the specific scope of our review. At the same time, the general statements expressed in the above-mentioned paragraph summarize the results of a vast number of single research articles and therefore we believe that a high quality, dedicated review like the one we have cited is a proper choice.

Line 45: The Authors should state that until to this date EMA has not approved the marketing authorization of Eteplirsen.

This statement was already present in our original manuscript; we have modified its wording to make it more clear.

Line 48: The Authors need to cite this information. "When it comes to generalized clinical practice, the only available option is to try and delay the progression of the disease by using either of two glucocorticoids, deflazacort or prednisone;"

We have added a reference to the relevant Guidelines for the treatment of DMD issued by the American Academy of Neurology.

Line 56: The Authors should correct the a.k.a. abbreviation.

Corrected.

Line 79: The Authors should be more careful to read the papers before cite the references.  Barnes 2016 does not cite the Nobel Prize won by Kendall, Reichstein and Hench for their discoveries relating to the hormones of the adrenal cortex, their structure and biological effects. It is relevant in a revision to have the correct citation.

We apologize for the mistake; the proper reference has been inserted in the revised version.

Lines 94-97: The Authors should use a reference for this paragraph. "Side effects of glucocorticoids, especially occurring with long-term treatment, are well known across pediatric and adult populations. Different from adults, in which hypertension and glucose intolerance are frequent, in young boys the main issues are weight gain, behavioral disturbances, growth stunting, and osteoporosis. Gastrointestinal symptoms and cataracts are also quite common."

Once again, we have added a reference to the relevant Guidelines for the treatment of DMD issued by the American Academy of Neurology.

Line 267: The Authors should change "the Authors" for Kobayashi et al.

Corrected.

Line 286: If the Authors would like to propose a discussion around flavocoxid, even that isn't approved by FDA anymore, the Authors should consider to discuss flavocoxid in the DMD aspect, which is the aim of the revision. In this case, the Authors should include and discuss the following reference: Flavocoxid counteracts muscle necrosis and improves functional properties in mdx mice: a comparison study with methylprednisolone. Messina S1, Bitto A, Aguennouz M, Mazzeo A, Migliorato A, Polito F, Irrera N, Altavilla D, Vita GL, Russo M, Naro A, De Pasquale MG, Rizzuto E, Musarò A, Squadrito F, Vita G. Exp Neurol. 2009 Dec;220(2):349-58. doi: 10.1016/j.expneurol.2009.09.015. Epub 2009 Sep 25.

We had decided to discuss Flavocoxid despite its recent withdrawal from commercialization because we felt that the topic of food supplements is still one of interest for the DMD community. The reference mentioned by the reviewer was indeed present in our manuscript, and was further commented to describe the effects of Flavocoxid in DMD.

Line 295-313: The Authors should include a discussion of the data obtained in mdx cardiac and skeletal muscles using Aminoglycosides.  Some suggestions are:

Aminoglycoside antibiotics restore dystrophin function to skeletal muscles of mdx mice. Barton-Davis ER, Cordier L, Shoturma DI, Leland SE, Sweeney HL. J Clin Invest. 1999 Aug;104(4):375-81.

Aminoglycoside Enhances the Delivery of Antisense Morpholino Oligonucleotides In Vitro and in mdx Mice. Wang M, Wu B, Shah SN, Lu P, Lu Q. Mol Ther Nucleic Acids. 2019 Jun 7;16:663-674. doi: 10.1016/j.omtn.2019.04.023. Epub 2019 May 2.

The Authors should also include in the discussion other reading through drugs such as RTC13.

Read-through compound 13 restores dystrophin expression and improves muscle function in the mdx mouse model for Duchenne muscular dystrophy. Kayali R, Ku JM, Khitrov G, Jung ME, Prikhodko O, Bertoni C. Hum Mol Genet. 2012 Sep 15;21(18):4007-20. doi: 10.1093/hmg/dds223. Epub 2012 Jun 12.

The section regarding the use of aminoglycosides in DMD has been rewritten/expanded, also in view of the comments by Reviewer 3, and now includes several other references, amongst which are also those mentioned by the Reviewer.

Lines 315-322: In order to understand the relevance of the drugs discussed in this chapter, the Authors should include clinical parameters that are influenced or modified in the DCM, such as echo, ECK and histological parameters.

We appreciated this Reviewer suggestion and understand its rationale. However, an in-depth discussion of DCM outcomes and parameters would make this review too long and less focused, especially when considering that a similar discussion of motor outcomes has not been included. However, we did expand the specific section in order to mention the most relevant cardiac outcomes and have provided a specific reference that addresses this specific topic.

Line 326: The Authors should include a reference for this statement. “Angiotensin-converting enzyme inhibitors (ACEi) are probably the most frequently prescribed cardiac medications in DMD. By inhibiting the renin-angiotensin pathway, they reduce peripheral circulatory resistance and blood volume (through a reduction of mineralocorticoid secretion).”

We have added a reference to the latest version of the guidelines issued by the DMD Care Considerations Working Group.

Line 333: The Authors should keep the abbreviation initiated on line 326 (ACEi).

Corrected.

Line339: The plural of ACEI is already included in the abbreviation and it is not necessary to add another "s"

Corrected.

Line 359: The Authors should provide data to support the usefulness of sacubitril/ valsartan.

We have clarified in the revised version that sacubitril/valsartan has not been specifically studied in DMD, and therefore no evidence is available; it has been mentioned as an interesting, potential candidate for repurposing.

Reviewer 3 Report

In this article the authors clearly describe many attempts at repurposing pharmaceuticals for MD. The topic is timely and there is need for a review such as this. I have one major concerns with the manuscript, and a number of grammatical, small changes required.

Major- There is quite a bit more data on the Ataluren story. This deserves a few more sentences.

Minor-

line 119 should read 'effects are surely crucial mechanisms (Figure 1' 145 skeletal muscle dysfunction and pathologic histology... 151 is …(MAO) - this statement needs to be referenced 167 dysfunction in muscle cells 174 Such effects have been 205 before the appearance of 206 which is being utilized for age-adjusted declines in recent 232 inhibits 238 long-term beneficial effects (this is a suggestion) 265 delete functional 267 authors 283 delete , all of which (this is a suggestion) 359 usefulness 375 Last 376 for all of these 383 in the foreseeable

Author Response

In this article the authors clearly describe many attempts at repurposing pharmaceuticals for MD. The topic is timely and there is need for a review such as this. I have one major concerns with the manuscript, and a number of grammatical, small changes required.
Major- There is quite a bit more data on the Ataluren story. This deserves a few more sentences.

We thank the Reviewer for his/her comments and careful text reviewing. As suggested, we have rewritten/expanded the section regarding the use of aminoglycosides in DMD. Although we agree with the Reviewer that Ataluren is a very interesting molecule in terms of DMD treatment, we mentioned it solely because its origin came from the same therapeutic approach pursued with the aminoglycosides; however, as it has been specifically designed for DMD we feel that it does not qualify as an example of drug repurposing.

All the minor comments were addressed in the revised manuscript.